# InterACT: Inter-dependency Aware Action Chunking with Hierarchical Attention Transformers for Bimanual Manipulation

**Andrew Lee**[1]    **Ian Chuang**[1,2]    **Ling-Yuan Chen**[1]    **Iman Soltani**[1]
[1] University of California Davis    [2] University of California Berkeley

**Abstract:** Bimanual manipulation presents unique challenges compared to unimanual tasks due to the complexity of coordinating two robotic arms. In this paper, we introduce InterACT: Inter-dependency aware Action Chunking with Hierarchical Attention Transformers, a novel imitation learning framework designed specifically for bimanual manipulation. InterACT leverages hierarchical attention mechanisms to effectively capture inter-dependencies between dual-arm joint states and visual inputs. The framework comprises a Hierarchical Attention Encoder, which processes multi-modal inputs through segment-wise and cross-segment attention mechanisms, and a Multi-arm Decoder that generates each arm's action predictions in parallel, while sharing information between the arms through synchronization blocks by providing the other arm's intermediate output as context. Our experiments, conducted on various simulated and real-world bimanual manipulation tasks, demonstrate that InterACT outperforms existing methods. Detailed ablation studies further validate the significance of key components, including the impact of CLS tokens, cross-segment encoders, and synchronization blocks on task performance. We provide supplementary materials and videos on our project page[1].

**Keywords:** Bimanual Manipulation, Imitation Learning

## 1 Introduction

Bimanual manipulation tasks, such as unscrewing a bottle cap or connecting two electrical cables, presents significant challenges due to the need for high precision and complex coordination between the two arms. Traditional approaches often rely on high-end robots and precise sensors, which can be expensive and require meticulous calibration [1, 2, 3]. However, recent advances in learning-based approaches offer the potential to perform such complex tasks using low-cost hardware. The ALOHA and the Action Chunking with Transformers (ACT) framework has shown that low-cost systems can achieve high-precision tasks that were traditionally only possible with expensive setups. The ACT addresses compounding error problem in imitation learning by predicting sequences of actions rather than single steps, thereby reducing the task's effective horizon and mitigating errors over time [4].

Despite recent advances, achieving the complex coordination between both arms required for consistent and successful execution of even relatively simple tasks remains a challenge in bimanual robotics. In this work, we propose InterACT: a new policy designed for bimanual manipulation that emphasizes inter-dependencies between two arms by utilizing hierarchical attention mechanisms. In our designs, multimodal inputs are encoded through segment-wise and cross-segment encoders which handle the complex relationships between different segments in a manner similar to how long documents are processed in NLP [5]. This combines the proprioceptive data of the robot arm joints and the visual features of the camera in a coherent latent space that allows for coordinated detail-oriented and smooth action execution.

---

[1] https://soltanilara.github.io/interact/

8th Conference on Robot Learning (CoRL 2024), Munich, Germany.

Our method advances beyond existing approaches by emphasizing the hierarchical structure of bimanual tasks: while independent arm movements are managed at the lower level, coordination occurs at a higher level, allowing each arm's actions to be informed by the other's movements as well as the environmental sensory feedback.. Leveraging the latest advances in attention mechanisms, this work presents a novel solution for bimanual manipulation tasks.

## 2 Related Works

### 2.1 Bimanual Manipulation

Bimanual manipulation involves two robotic arms performing tasks that require dexterity and coordination. Inspired by the natural ability of humans to perform such tasks, researchers have been keen on modeling these skills in robots. Prevailing methodologies, including classical control methods, reinforcement learning, and imitation learning, have significantly advanced the field.

Early research primarily relied on classical control methods, focusing on predefined trajectories and high-fidelity models to achieve coordinated movements [1, 2, 3]. However, these methods often required extensive calibration and were less adaptable to dynamic environments. The introduction of reinforcement learning (RL) made bimanual manipulation more adaptable and robust. RL-based approaches have proven effective in handling complex tasks, outperforming classical methods, and generalizing across different scenarios [6, 7, 8, 6, 9]. These methods leverage RL's ability to learn from interactions with the environment, improving performance in varied and unpredictable conditions [10, 11, 12]. Imitation learning has emerged as a prominent method for teaching robots tasks through human demonstrations. This approach enables robots to mimic complex human actions, facilitating the execution of intricate tasks [13, 14]. Research has demonstrated the effectiveness of imitation learning in training robots for coordinated dual-arm manipulation leveraging waypoints, hierarchical skill learning and force-based techniques [15, 16, 17, 18, 19, 20]. Frameworks like ALOHA have shown that low-cost systems can perform high-precision tasks using imitation learning techniques traditionally reserved for expensive setups [4, 21]. Similarly, the Action Chunking with Transformers (ACT) algorithm addresses the compounding error problem in imitation learning by predicting sequences of actions, improving accuracy and efficiency [4].

Despite these advancements, bimanual manipulation remains challenging. Recent research has explored stabilizing one arm [22], simplifying actions [23], integrating additional multi-modal data, such as language or sensory feedback [24, 25, 26], to enhance the robustness and efficiency of bimanual manipulation systems. These efforts hold promise for developing more adaptable robotic systems capable of performing more complex manipulation tasks.

### 2.2 Hierarchical Attention Mechanisms

Hierarchical attention [27] mechanisms have gained prominence for their ability to process and integrate multi-modal inputs. These mechanisms aggregate information at multiple levels of granularity, making them well-suited for tasks requiring both local and global context understanding [28, 29].

Hierarchical attention has shown considerable success in other domains, such as natural language processing (NLP), where hierarchical attention transformers have been effectively used for long document classification [5, 30, 31]. These models focus on different parts of the input text at varying levels of abstraction, enhancing the ability to handle long and complex documents. Extensions of foundational attention mechanisms [32], such as the Hierarchical Attention Network (HAN) [27] and hierarchical representations in BERT [33], further demonstrate their potential in managing multi-layered information.

In robotic manipulation, the concept of hierarchy has been explored to manage the complexity of long-horizon tasks by breaking down problems into manageable sub-tasks [34], but not at the policy's attention layer level. As proven in long document classification, leveraging segment-wise and cross-segment attention mechanisms, hierarchical attention models can capture dependencies within

and across sentences [5]. This makes hierarchical attention transformers particularly appealing for use in bimanual robotic manipulation tasks, where capturing dependencies between arms is crucial.

In this work, we tailor the transformer architecture featuring hierarchical attention to the bimanual robotics tasks and explore its utility in extracting the complex inter-dependencies between the actions of the two arms. We hypothesize that this can lead to more coordinated actions and hence, more robust performance.

# 3 InterACT: Inter-dependency Aware Action Chunking with Hierarchical Attention Transformer

The ACT model [4] leverages transformer architecture to predict future steps in bimanual manipulation tasks, effectively handling sequences of actions by capturing temporal dependencies. However, it does not explicitly model inter-dependencies between dual-arm joint states and visual inputs, which can limit its performance in complex manipulation tasks.

InterACT builds upon the ACT model, enhancing it with hierarchical attention mechanisms to capture inter-dependencies between dual-arm joint states and visual inputs. This section provides an overview of the InterACT model and its key components: the **Hierarchical Attention Encoder**, which processes multi-modal inputs to capture both intra- (corresponding to one arm or sensory input) and inter-segment (across arms or sensory inputs) dependencies, and the **Multi-arm Decoder**, which generates synchronized action predictions for both arms.

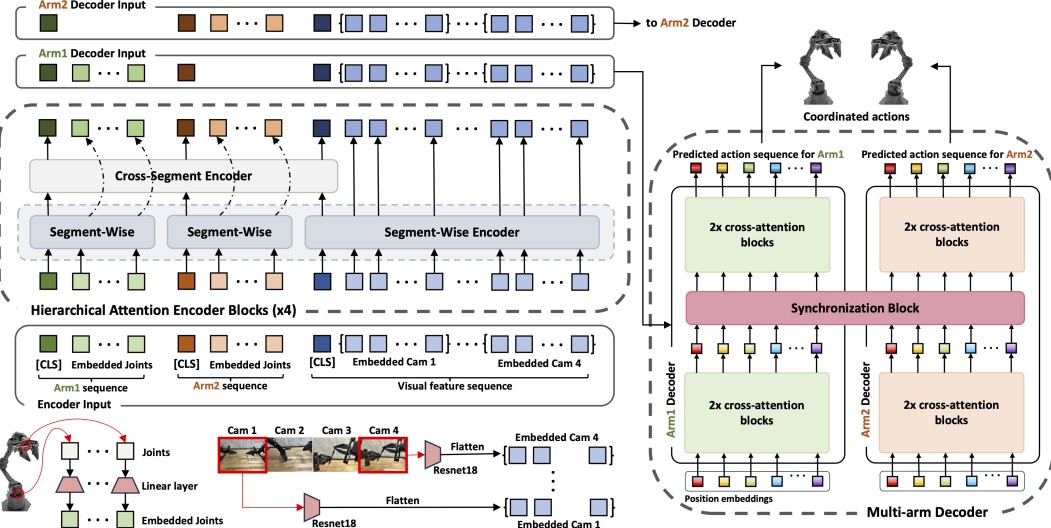

Figure 1: **Architecture of the InterACT**. The Hierarchical Attention Encoder consists of multiple blocks of segment-wise encoders and cross-segment encoder. The output is passed through the Multi-arm Decoder which consists of Arm1 and Arm2 specific decoders that process the input segments independently. The synchronization block allows for information sharing between the two decoders.

## 3.1 Hierarchical Attention Encoder

Segments are defined as independent groups of data inputs that are processed individually before integration. In this context, segments include the joint states of each arm and visual features at a specific timestep. The Hierarchical Attention Encoder processes input segments and captures both intra-segment and inter-segment dependencies through a hierarchical attention mechanism that consists of two primary components: the Segment-wise encoder and the Cross-segment encoder. A detailed pipeline for the Hierarchical Attention Encoder is illustrated in Figure 1 and Algorithm 1.

**Algorithm 1:** Hierarchical Attention Encoder

---

1: Given: Demo dataset $\mathcal{D}$, segment-wise encoder $E_{\text{seg}}$, cross-segment encoder $E_{\text{cross}}$, CLS tokens CLS, number of layers $L$.
2: Let $S_i$ represent the input segment at index $i$, $\text{CLS}_i$ represent the CLS tokens for segment $i$, and $S_{\text{visual}}$ represent the visual features.
3: Initialize segment-wise encoder $E_{\text{seg}}$
4: Initialize cross-segment encoder $E_{\text{cross}}$
5: **for** each segment $S_i$ in $\mathcal{D}$ **do**
6:     Prepend CLS tokens: $S'_i \leftarrow [\text{CLS}_i, S_i]$
7:     Add positional encoding to $S'_i$
8:     **for** each layer $l = 1, 2, ..., L$ **do**
9:         $S'_i \leftarrow E_{\text{seg}}(S'_i)$
10:     **end for**
11: **end for**
12: Extract CLS tokens: $\text{CLS}_{\text{tokens}} \leftarrow [\text{CLS}_1, \text{CLS}_2, ..., \text{CLS}_N]$
13: Add positional encoding to $\text{CLS}_{\text{tokens}}$
14: **for** each layer $l = 1, 2, ..., L$ **do**
15:     $\text{CLS}_{\text{tokens}} \leftarrow E_{\text{cross}}(\text{CLS}_{\text{tokens}})$
16: **end for**
17: Return final encoded states $\{\text{CLS}_{\text{arm1}}, S'_{\text{arm1}}, \text{CLS}_{\text{arm2}}, S'_{\text{arm2}}, \text{CLS}_{\text{visual}}, S'_{\text{visual}}\}$

---

Each raw joint position is embedded into a single token through a linear layer. Visual features are extracted from RGB images using ResNet18 backbones, which convert and flatten the images along the spatial dimension to form a sequence of visual feature tokens. The joint sequences from both arms and the visual feature sequence are then concatenated to form a combined input sequence. Classification tokens (CLS tokens) are prepended to each segment, allowing the model to capture and summarize segment information during attention [33, 35]. Positional embeddings are applied to the sequence to ensure the model can understand the order of the tokens within each segment sequence.

**Segment-Wise encoder** is responsible for capturing intra-segment dependencies by processing each segment individually. Each segment represents a distinct modality, and the encoder's task is to aggregate the information within each segment. The self-attention mechanism allows each token within the segment to attend to every other token, including the CLS tokens, thereby allowing the CLS token to capture intra-segment dependencies. This aggregation of information enables the model to effectively leverage the relationships within different parts of the segment [5], such as the dependencies between joints or the spatial relationships within visual inputs.

**Cross-segment encoder** is responsible for capturing inter-segment dependencies by handling the CLS tokens generated from each segment by the segment-wise encoder. The CLS tokens serve as a condensed representation of each segment, summarizing the key information within the segment. By focusing on these CLS tokens, the cross-segment encoder reduces the computational overhead while effectively integrating information across different modalities.

Multiple segment-wise and cross-segment encoders are stacked, allowing the model to progressively refine its understanding of both intra- and inter-segment relationships. We follow an interleaved stacking approach [5], where segment-wise and cross-segment encoders alternate. This deep stacking enables the model to capture the complex dependencies across segments.

### 3.2 Multi-arm Decoder

Similar to multi-task learning in NLP [36, 37], the Multi-arm Decoder comprises two parallel paths of decoder blocks, each dedicated to processing the encoded states and target tokens to generate the predicted actions for one of the arms. This separation makes sense in bimanual manipulation tasks because, while the arms must coordinate, they often perform independent actions that require individual attention and control. By using separate decoders, each arm can process its specific actions while still sharing critical information through synchronization mechanisms.

This section details the components and workflow of the Multi-arm Decoder, highlighting how it employs enriched input sequence and CLS tokens from the encoder to coordinate actions between both arms effectively. A detailed pipeline for the Multi-arm Decoder is illustrated in Figure 1 and Algorithm 2.

---

**Algorithm 2:** Multi-arm Decoder

---

1: Given: Target tokens with positional embeddings $T_{\text{pos}}$, number of layers $L$.
2: Let $D_{\text{arm1}}$, $D_{\text{arm2}}$ represent decoders for Arm1 and Arm2 respectively.
3: Let $D_{\text{sync}}$ represent Synchronization block.
4: Initialize cross-attention blocks for $D_{\text{arm1}}$ and $D_{\text{arm2}}$
5: Initialize synchronization block (multihead self-attention)
6: **Arm1 Specific Decoder:**
7: **for** each layer $l = 1, 2, ..., L$ **do**
8:     $Arm1_{\text{input}} \leftarrow \{S_{\text{arm1}}, \text{CLS}_{\text{arm2}}, \text{CLS}_{\text{visual}}, S_{\text{visual}}\}$
9:     $Arm1_{\text{output}} \leftarrow D_{\text{arm1}}(Arm1_{\text{input}}, T_{\text{pos}})$
10: **end for**
11: **Arm2 Specific Decoder:**
12: **for** each layer $l = 1, 2, ..., L$ **do**
13:     $Arm2_{\text{input}} \leftarrow \{\text{CLS}_{\text{arm1}}, S_{\text{arm2}}, \text{CLS}_{\text{visual}}, S_{\text{visual}}\}$
14:     $Arm2_{\text{output}} \leftarrow D_{\text{arm2}}(Arm2_{\text{input}}, T_{\text{pos}})$
15: **end for**
16: **Synchronization Block:**
17: $Concatenated_{\text{output}} \leftarrow \{Arm1_{\text{output}}, Arm2_{\text{output}}\}$
18: $Shared_{\text{output}} \leftarrow D_{\text{sync}}(Concatenated_{\text{output}})$
19: **Split Shared Output:**
20: $Shared_{\text{output\_arm1}}, Shared_{\text{output\_arm2}} \leftarrow \text{Split}(Shared_{\text{output}})$
21: **Arm Specific Decoder**
22: $Arm1_{\text{final\_output}} \leftarrow D_{\text{arm1}}(Arm1_{\text{input}}, Shared_{\text{output\_arm1}})$
23: $Arm2_{\text{final\_output}} \leftarrow D_{\text{arm2}}(Arm2_{\text{input}}, Shared_{\text{output\_arm2}})$
24: Return $Arm1_{\text{final\_output}}, Arm2_{\text{final\_output}}$

---

Encoded states from the Hierarchical Attention Encoder are used as the context for decoding. Target tokens, initialized with positional embeddings, serve as the starting point for generating sequence of actions. The input for each decoder includes relevant segments and CLS tokens to facilitate cross-attention mechanisms.

**Arm Specific Decoders** are responsible for generating intermediate decoder outputs for each arm. The Arm1 Decoder takes in Arm1 segments, Arm2 CLS tokens, and visual feature segments, while the Arm2 Decoder takes in Arm2 segments, Arm1 CLS tokens, and visual feature segments. Each layer processes these inputs through a cross-attention mechanism with the target tokens, incorporating contextual information from both joint states and visual features. This design enables each arm to use contextual information from the other arm ensuring synchronized actions.

**Synchronization Block** enhances coordination between both arms. Before generating the final output from each decoder, the intermediate outputs from both decoders are concatenated and processed through a synchronization block, which uses self-attention to integrate shared information. This step ensures that both arms leverage the combined context from the other arm and visual inputs before passing through the reset of the decoder layers. The use of attention mechanisms for sharing information across multiple decoders has also been explored in multi-task learning, demonstrating its effectiveness in improving performance and coherence within tasks [38].

### 3.3 Training and Evaluation

InterACT is trained using an end-to-end imitation learning framework adapted from the ACT algorithm. The training process involves collecting high-quality human demonstrations through a teleoperation system, capturing joint positions and RGB images at 50Hz. The collected data is preprocessed to extract joint states and visual features using ResNet18 backbones, converting the RGB images into feature tokens. Each joint state and visual feature is tokenized, with multiple CLS tokens prepended to summarize each segment's information. Positional embeddings are added to retain se-

quence information. Action chunking is implemented to predict sequences of actions rather than single steps, reducing the task's effective horizon and mitigating compounding errors. Additionally, a temporal ensemble method is employed to improve the temporal consistency and robustness of the action predictions by weighing predictions over multiple time steps [4].

Evaluation is conducted on both simulated and real-world tasks, measuring success rates to assess the model's performance in generating accurate and coordinated actions.

## 4 Experiments and Results

For the real-robot setup, we modified the ALOHA 2 [39] setup by adjusting the height of the top camera to improve the visibility of the tabletop environment. This adjustment ensures that the camera captures a more comprehensive view of the workspace, which is essential for accurately tracking the bimanual manipulation tasks. Our robot setup is illustrated in Appendix B.

To evaluate our model, we conducted experiments on three simulation tasks and six real-world tasks: **Transfer Cube** and **Peg Insertion** along with **Slide Ziploc** and **Thread Velcro** are tasks adapted from ACT [4]. We also introduce five new tasks: one simulation task and four real-world tasks. The simulation task is **Slot Insertion**, where both arms need to grab each side of a long peg together and place it in a slot on the table. The new real-world tasks include **Insert Plug**, **Click Pen**, **Sweep**, and **Unscrew Cap**. We collected 50 demonstrations for each task. For the simulation tasks, the data used to train the model were all from human demonstrations and we did not evaluate performance of the models trained on scripted data. Detailed task definitions are provided in Appendix A, and the hyperparameters of both ACT and InterACT are provided in Appendix C.

To Evaluate the performance of InterACT, we chose to compare our model against ACT [4] and the Diffusion policy [40]. However, with only 50 demonstrations, the Diffusion policy failed to execute any of the nine tasks. We did not perform any additional tuning on the Diffusion policy, as its poor performance suggested it was not well-suited for tasks with limited demonstration data. Therefore, we focus our comparisons exclusively on ACT, as ACT already significantly outperforms several other policies, including BC-ConvMLP [41, 14], BeT [42], and VINN [43], by a large margin in bimanual manipulation tasks [4].

| | Transfer Cube (Sim) | | | Peg Insertion (Sim) | | | Slide Ziploc (Real) | | | Thread Velcro (Real) | | |
|---|---|---|---|---|---|---|---|---|---|---|---|---|
| | Touch | Lift | **Transfer** | Grasp | Contact | **Insert** | Grasp | Pinch | **Open** | Lift | Grasp | **Insert** |
| ACT | 82 | 60 | 50 | 76 | 66 | 20 | 96 | 92 | 88 | 88 | 42 | 16 |
| **InterACT** | **98** | **88** | **82** | **88** | **78** | **44** | 96 | 92 | **92** | **94** | **56** | **20** |

| | Slot insertion (Sim) | | Insert Plug (Real) | | Click Pen (Real) | | Sweep (Real) | | Unscrew cap (Real) | |
|---|---|---|---|---|---|---|---|---|---|---|
| | Lift | **Insert** | Grasp | **Insert** | Grasp | **Click** | Grasp | **Sweep** | Touch | **Unscrew** |
| ACT | 96 | 88 | 92 | 30 | 92 | 56 | 88 | 42 | 84 | 60 |
| **InterACT)** | **100** | **100** | 92 | **42** | **94** | **62** | **92** | **52** | **88** | **62** |

Table 1: Success rate (%) for tasks adapted from ACT [4] (top) and our original tasks (bottom). For simulation tasks, we averaged the results across 3 random seeds over 50 episodes each. The real-world tasks were also evaluated over 50 episodes.

### 4.1 Results

The results of our experiments are summarized in Tables 1. Our InterACT model shows superior performance compared to ACT on all simulated and real-world tasks. In the simulated tasks, InterACT outperformed ACT significantly, particularly in the *Transfer Cube* and *Peg Insertion* tasks where coordination and precision are crucial. The success rates for the "Transfer" stages in the *Transfer Cube* task, as well as the "Insert" stages in the *Peg Insertion* task, were notably higher with InterACT, demonstrating the effectiveness of our method in tasks that require coordination between the two arms. Moreover, in newly introduced tasks such as *Slot Insertion* and *Insert Plug*, InterACT also demonstrated higher performance. The *Slot Insertion* task, which requires precise coordination

| | Transfer Cube | | | Peg Insertion | | | Slot Insertion | |
|---|---|---|---|---|---|---|---|---|
| | Touch | Lift | **Transfer** | Grasp | Contact | **Insert** | **Lift** | Insert |
| InterACT (no CLS Tokens) | 98 | 84 | 84 | 70 | 68 | 22 | 100 | 86 |
| InterACT (no CS Encoder) | 80 | 72 | 72 | 84 | 80 | 24 | 100 | 98 |
| InterACT (no Sync Block) | 74 | 54 | 54 | **90** | **86** | 30 | 100 | 100 |
| **InterACT (all components)** | **98** | **88** | **84** | 88 | 78 | **44** | **100** | **100** |

Table 2: Success rate (%) for simulation tasks under different conditions, InterACT model with InterACT model without CLS tokens, cross-segment (CS) encoder, and synchronization block. Coordination subtasks are indicated in bold.

between two arms to carry the peg and adjust for alignment, showed a 100% success rate with InterACT, compared to 88% with ACT. Similarly, in the *Insert Plug* task, InterACT achieved better results in the coordination subtask "Insert". This highlights the robustness of our model in handling tasks that require precise coordination between the two arms.

Overall, the experimental results validate the effectiveness of our hierarchical attention framework. By improving coordination and precision in bimanual manipulation tasks, our InterACT model provides a more robust solution for complex bimanual manipulation challenges in both simulation and real-world scenarios.

### 4.2 Ablation Studies

In this section, we perform ablation studies to evaluate the contributions of different components of the InterACT framework. Specifically, we focus on the impact of CLS tokens, the cross-segment encoder, and the synchronization block in the decoder.

**Impact of CLS Tokens:** To assess the impact of CLS tokens, we conducted experiments with and without CLS tokens as input to the decoder. The results, summarized in Table 2, showed no significant difference in the easier *Transfer Cube* task. However, there were notable improvements in the success rates of the more complex *Peg Insertion* task when CLS tokens were included. The aggregated information in the CLS tokens enhances the model's ability to generate accurate and coordinated actions, particularly in tasks requiring higher precision and synchronization.

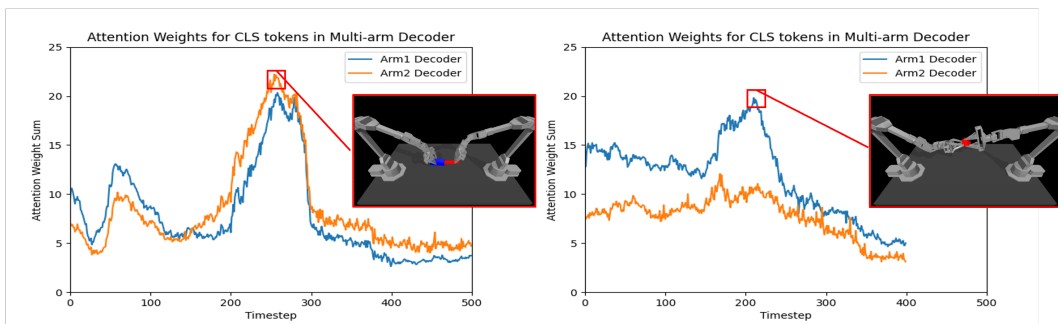

Figure 2: **Attention weights for CLS tokens at the Multi-arm Decoder over time for *Peg Insertion* (right) and *Transfer Cube* (right).** The red highlighted sections correspond to specific timesteps in executing the task. Spikes in attention weights are observed during coordinated phase.

To gain deeper insights into the model's behavior, we studied how attention weights to CLS tokens at the decoder change over the timesteps. The results illustrated in Figure 2 showed that during the phase of interaction between the two arms, significant spikes in attention weights were observed. These spikes occurred at key moments where coordinated actions between the arms were necessary. This indicates that the model heavily relies on the CLS tokens to process information when coordinating actions between the arms. This observation highlights the CLS tokens' importance in facilitating precise bimanual manipulation.

**Impact of Cross-segment Encoder:** The cross-segment encoder captures inter-segment dependencies, allowing the model to effectively integrate information from different joints across the two arms as well as the camera frames. The results indicate that removing the cross-segment encoder significantly decreases performance in complex tasks. For example, in the *Slot Insertion* task, the success rate for the coordination subtask "Insert" dropped to 24% from 44% without the cross-segment encoder. This highlights the importance of capturing inter-segment dependencies for generating accurate and coordinated actions.

**Impact of Synchronization Block:** The synchronization block enhances coordination between the two arms by sharing contextual information at the decoder.. This is crucial for synchronized and efficient bimanual manipulation. The results show that the removal of the synchronization block leads to a significant drop in performance across all tasks, particularly in the *Transfer Cube* and *Peg Insertion* tasks. This demonstrates the necessity of the synchronization block for achieving coordinated and actions.

Our ablation studies clearly illustrate that all components of the proposed InterACT framework—CLS tokens, cross-segment encoder, and synchronization block—play a critical role in achieving high success rates in coordination tasks. The best performance for the coordination subtask is achieved when all components are utilized, underscoring the importance of the holistic integration of these elements in the InterACT framework.

## 5 Conclusion and Future Work

In this work, we introduced InterACT, a framework for robust bimanual manipulation, which integrates hierarchical attention transformers to capture inter-dependencies between dual-arm joint states and visual inputs. The key contributions of our work include the development of a Hierarchical Attention Encoder and a Multi-arm Decoder. The Hierarchical Attention Encoder aggregates intra-segment information using a segment-wise encoder and integrates inter-segment dependencies through a cross-segment encoder. The Multi-arm Decoder, while generating action sequence for each arm in parallel, ensures coordinated action sequence generation through synchronization blocks. Our experiment results on both simulated and real-world tasks demonstrate the superior performance of InterACT compared to the baseline ACT. The ablation studies further highlight the importance of these components in our framework. The use of CLS tokens, cross-segment encoder and the synchronization block significantly enhances the model's ability to generate accurate and coordinated actions, leading to higher success rates in bimanual manipulation tasks.

One of the limitations of InterACT is that the number of CLS tokens and the number of encoder blocks are influenced by the level of coordination required for a given task. While we set a fixed number of CLS tokens across all tasks, finding the optimal hyperparameter requires heavy tuning, as there is no clear rule to guide this selection. Additionally, the level of coordination between the two arms is not straightforward to measure. Coordination can depend on task-specific factors such as temporal synchronization, dependency between arm movements, and precision, which are not easily quantifiable. As a result, determining the optimal hyperparameters becomes even more challenging, requiring manual adjustments and multiple experiments to ensure optimal performance. One potential approach to mitigate this challenge could involve adding an auxiliary loss term that explicitly encourages coordinated actions between the two arms, which might help improve coordination without relying on extensive tuning. However, this may require quantitative definition of coordination and further experimentation. As it stands, the scalability of our approach to more complex tasks or scenarios with diverse coordination requirements remains a challenge.

While InterACT has shown promising directions in integrating robotic arm joints and visual inputs, it has not yet explored integration with other modalities. Future work will explore integrating additional modalities, such as language or sensory feedback, to further improve the robustness of bimanual manipulation tasks. Additionally, addressing the hierarchical nature of tasks will be crucial for better task decomposition and execution. These enhancements could leverage the flexible attention mechanisms demonstrated in this work to manage the added complexity and data integration.

**Acknowledgments**

We thank the members of the UC Davis Laboratory for AI, Robotics, and Automation (LARA) for their valuable feedback, discussions, and assistance with data collection that contributed to this work.

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

# Appendix A: New task definitions

In this section, we define the new tasks we introduced in this work including one simulated task and four real-world tasks.

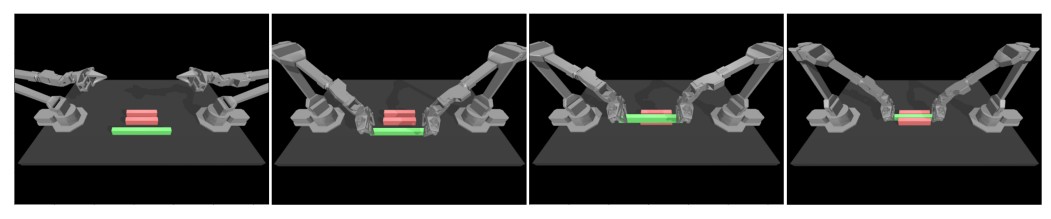

***Slot Insertion (Sim):*** Slot insertion is a simulated task where both arms need to lift each side of a long peg together (***Lift***) and place it in a slot on the table (***Insert***).

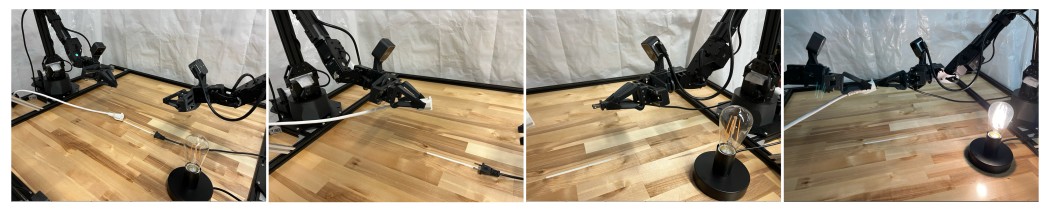

***Insert Plug (Real):*** Insert Plug is a real-world task where each arm grabs a male and a female electrical plug respectively (***Grasp***) and connects the two plugs above the table (***Insert***).

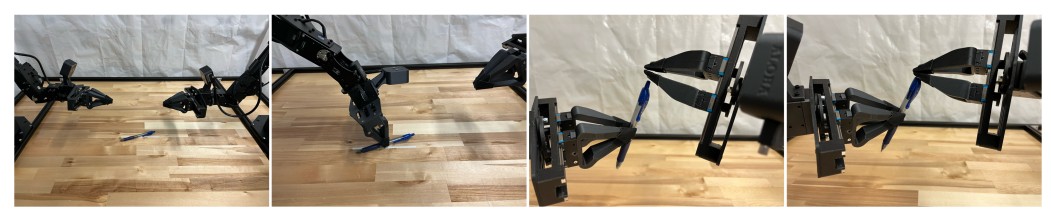

***Click Pen (Real):*** Click Pen is a real-world task where each one (left) arm grabs a retractable pen in the middle (***Grasp***), and clicks the pen with the other (right) arm (***Click***).

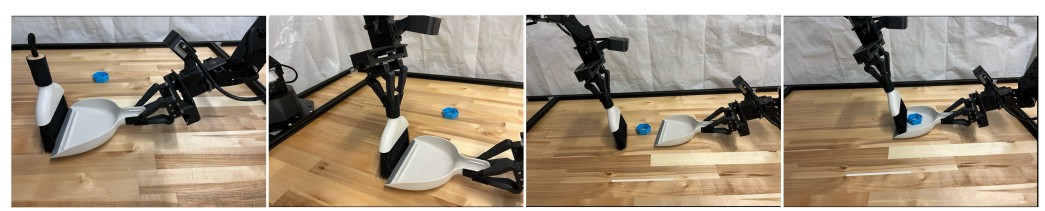

***Sweep (Real):*** Sweep is a real-world task where one arm grabs a brush and the other arm grabs a dustpan (***Grasp***). The arms then move towards a toy object lying on the table and sweep it into the dustpan (***Sweep***).

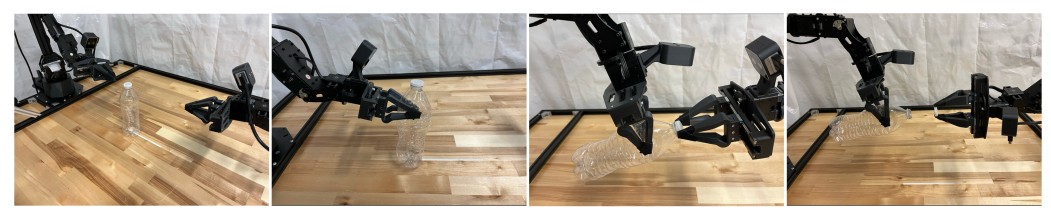

***Unscrew Cap (Real):*** Unscrew Cap is a real-world task where one arm grabs a plastic water bottle while the other arm reaches and touches the bottle cap (***Touch***), then grabs the bottle cap and unscrews it (***Unscrew***).

## Appendix B: Real-robot Setup

We utilize the ALOHA 2 setup [39] for our real-world experiments. Rather than cropping the top camera frame, we lower the camera's height to focus on the table environment, thereby maintaining resolution and capturing the necessary details. Additionally, similar to the ALOHA setup [4], we use a tarp around the setup to block unnecessary background distractions. These modifications help enhance the quality of data collected by ensuring that the attention is solely on the manipulation tasks. A photo of our setup is shown in Figure 3.

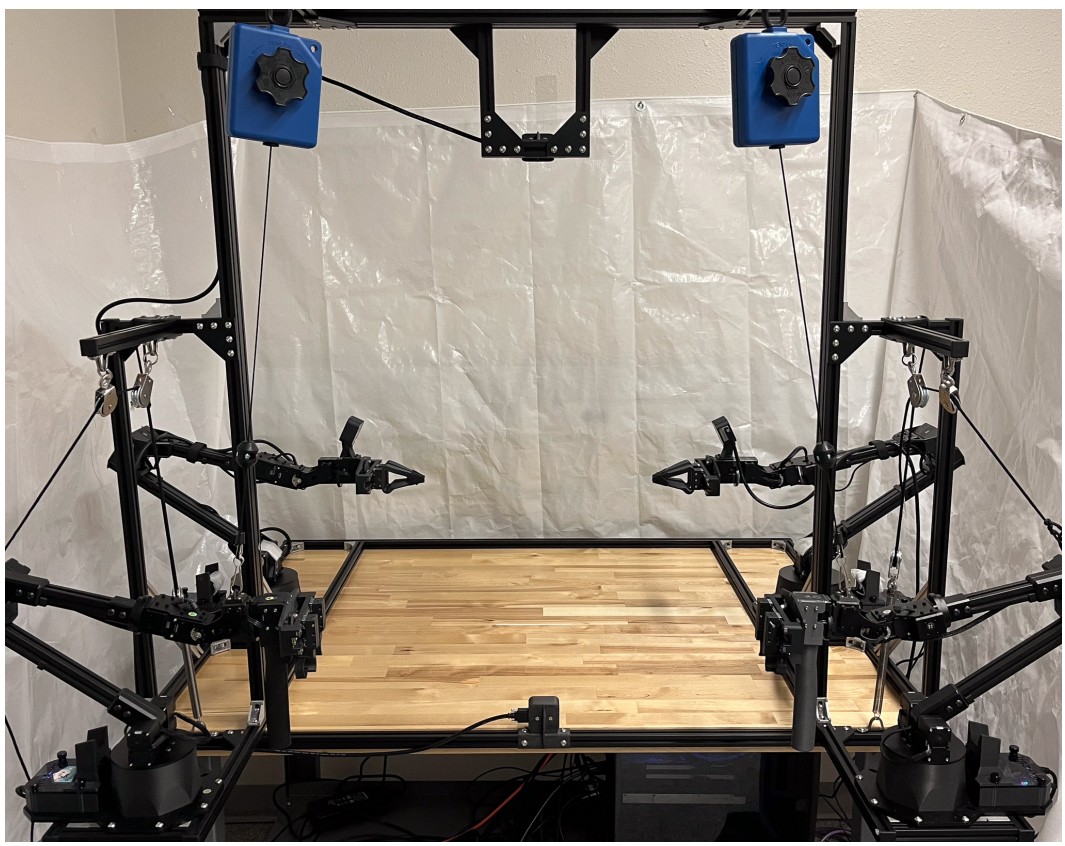

Figure 3: **Our Real-robot Setup**. We have modified the ALOHA 2 setup for our real-world experiments. Modifications include adjusting the camera height and using a tarp around the setup.

# Appendix C: Hyperparameters

In this section, we summarize the hyperparameters of InterACT and ACT models used for training and evaluation in this paper.

| Hyperparameters | |
|---|---|
| # Segment-Wise Encoder Layers | 3 |
| # Cross-Segment Encoder Layers | 3 |
| # Multi-arm Decoder Layers | 4 |
| # Synchronization Block Layers | 1 |
| # CLS tokens for Arm Joints | 7 |
| # CLS tokens for Visual Features | 5 |

Table 3: Unique hyperparameters of InterACT

| Hyperparameters | |
|---|---|
| # Encoder Layers | 4 |
| # Decoder Layers | 7 |

Table 4: Unique hyperparameters of ACT

| Hyperparameters | |
|---|---|
| Learning Rate | 1e-5 |
| Batch Size | 8 |
| Feedforward Dimension | 3200 |
| Hidden Dimension | 512 |
| # Heads | 8 |
| Chunk Size | 50 |
| Beta | 10 |
| Dropout | 0.1 |

Table 5: Common hyperparameters of InterACT and ACT

