# OpenReview forum: "InterACT: Inter-dependency Aware Action Chunking with Hierarchical Attention Transformers for Bimanual Manipulation"
_robot-learning.org/CoRL/2024/Conference — CoRL 2024_

### Official Review · Reviewer_Gcy9 · 2024-07-06
**good paper**

**Originality:** 3
**Technical Quality:** 3
**Clarity Of Presentation:** 4
**Potential Impact:** 4
**Recommendation:** 3
**Confidence:** 4

**Review:**

#### Strengths
- **Hierarchical Attention Mechanisms**: Effective in capturing inter-dependencies.
- **Multi-arm Coordination**: Addressing challenges specific to bimanual tasks.
- **Experimental Validation**: Robust testing across simulated and real-world scenarios.
- **Ablation Studies**: Detailed analysis of framework components.
- **Potential Impact**: Application in practical robotics domains.

#### Weaknesses
- **Complexity**: The effectiveness of hierarchical attention and synchronization blocks may introduce computational overhead, which could limit real-time application on less powerful hardware.
- **Limited Comparison**: The authors only compare with ACT. Although the author mentioned "We focus our comparisons exclusively on ACT as ACT already outperforms BC-ConvMLP [30, 31], BeT [32], and VINN [33] by a large margin in bimanual manipulation tasks [4]." It will be better to compare more potential baselines. For example, the reviewer found that "Hierarchical Attention Transformers" is also studied for general RL [1,2,3] and used in other fileds [4,5], which could be possible baselines.

[1] Transformer in Transformer as Backbone for Deep Reinforcement Learning. Arxiv 2022.
[2] PDiT: Interleaving Perception and Decision-making Transformers for Deep Rinforcement Learning. AAMAS 2024.
[3] Sequential Asynchronous Action Coordination in Multi-Agent Systems: A Stackelberg Decision Transformer Approach. ICML 2024.
[4] CoSLight: Co-optimizing Collaborator Selection and Decision-making to Enhance Traffic Signal Control. KDD 2024.
[5] X-Light: Cross-City Traffic Signal Control Using Transformer on Transformer as Meta Multi-Agent Reinforcement Learner. IJCAI 2024.

**Quality Of The Limitations Section:**

1

**Questions For Rebuttal:**

1. The Hierarchical Attention Transformers may introduce computational overhead, which could limit real-time application on real robot?

2. Could the authors provide more comparision with other baselines?

**Robotics Focus:**

2

**Summary Of Paper:**

The paper "InterACT" introduces a novel framework for bimanual manipulation using Hierarchical Attention Transformers (HAT). It addresses the challenge of coordinating actions between both arms by integrating segment-wise and cross-segment attention mechanisms. The framework includes a Hierarchical Attention Encoder for multi-modal input processing and a Multi-arm Decoder for synchronized action prediction. Key contributions include significant performance improvements over existing methods in various tasks, validated through extensive experiments and ablation studies that highlight the effectiveness of its components like CLS tokens and synchronization blocks.

**Summary Of Recommendation:**

Overall, InterACT appears to be a significant contribution to the field of robotic manipulation, leveraging advanced attention mechanisms to improve coordination and performance in bimanual tasks. Further peer review and empirical validation will be crucial for confirming its robustness and practical utility across various applications.

---

### Official Review · Reviewer_KWL4 · 2024-07-08
**The proposed method has shown noticeable performance improvements over the baseline; however, the novelty of this method may be limited.**

**Originality:** 2
**Technical Quality:** 2
**Clarity Of Presentation:** 3
**Potential Impact:** 2
**Recommendation:** 2
**Confidence:** 4

**Review:**

Strengths:
* The method is demonstrated on real-world, fine-grained bimanual manipulation tasks.
* The figures and algorithms are clear and helpful in understanding the method and results.


Weaknesses:
* Although the proposed method has shown noticeable performance improvements over the baseline, the novelty of this method is somewhat lacking since the idea of a hierarchical attention mechanism is not new.
* The proposed method is only being compared to one baseline (ACT), which may be limited. A baseline that could be included in the comparison is a bimanual Diffusion Policy.
* Many important details about the evaluation protocol are missing. Please refer to the “Questions for Rebuttal” section.


Minor weaknesses:
* The related works section is missing some recent papers on bimanual manipulation: [1, 2, 3, 4].
* Typo: “decoder” on line 128.


[1] J. Grannen, Y. Wu, B. Vu, and D. Sadigh. Stabilize to act: Learning to coordinate for bimanual manipulation. In Conference on Robot Learning (CoRL), 2023.

[2] L. X. Shi, A. Sharma, T. Z. Zhao, and C. Finn. Waypoint-based imitation learning for robotic manipulation. In Conference on Robot Learning (CoRL), 2023.

[3] A. Bahety, P. Mandikal, B. Abbatematteo, and R. Martín-Martín. ScrewMimic: Bimanual Imitation from Human Videos with Screw Space Projection. In Robotics: Science and Systems (RSS), 2024.

[4] J. Varley, S. Singh, D. Jain, K. Choromanski, A. Zeng, S. B. R. Chowdhury, A. Dubey, and V. Sindhwani. Embodied AI with Two Arms: Zero-shot Learning, Safety and Modularity. arXiv preprint arXiv:2404.03570, 2024.

**Quality Of The Limitations Section:**

1

**Questions For Rebuttal:**

My main concerns are discussed in the weaknesses section. Additionally, several important details about the evaluation protocol are missing:
* How many training seeds are used in the experiments?
* How many demonstrations are used to train the models?
* How many episodes are being evaluated?
* What are the environment variations?

**Robotics Focus:**

4

**Summary Of Paper:**

This paper presents a new imitation learning method for bimanual manipulation. Built on top of the Action Chunking with Transformers (ACT) framework, the method introduces a hierarchical attention encoder and a multi-arm decoder. Specifically, the encoder consists of segment-wise encoders and a cross-segment encoder to better capture the interdependencies between dual-arm joint states and visual inputs. The decoder includes arm-specific decoders and a synchronization block. The method is demonstrated on both simulated and real-world bimanual manipulation tasks.

**Summary Of Recommendation:**

I recommend a weak reject based on the issues discussed above. My primary concern lies in the novelty of the proposed method, as the idea of a hierarchical attention mechanism has been introduced previously. Additionally, a more comprehensive method comparison and the details of the evaluation protocol need to be included.

---

### Official Review · Reviewer_n8FP · 2024-07-20
**The research is well-executed but needs clarification on marginal improvement over ACT**

**Originality:** 4
**Technical Quality:** 4
**Clarity Of Presentation:** 5
**Potential Impact:** 3
**Recommendation:** 3
**Confidence:** 4

**Review:**

The research is well-executed, my major concern is the marginal improvement compared to ACT if trained with scripted data (See question 1). It would be good if the authors can clarify this point if the numbers have been copied from the original ACT work and how the new method was trained.

Other than that, writing can be improved by adding numbers and supporting some claims made in this work.
While the related works section is comprehensive, it could be further improved by including a discussion of other bimanual manipulation methods beyond ACT. Many existing methods, although often tailored to specific tasks, could provide a broader context for the contributions of this work.

The figures in the paper could be revised for clarity, and additional figures could be added to provide more visual insight into the results and methodology.


### Minor:

- Line 36, 210: Seems to be generated by GPT. Some additional information Is missing. E.g. citations or numbers, e.g. the relative improvement compare to ACT?
- Line 252: synchronizatin

**Quality Of The Limitations Section:**

2

**Questions For Rebuttal:**

1.)	ACT reports different numbers if training with scripted data. It seems  that this work has copied the numbers when training with human data from ACT. Are the experiments for InterACT scripted or trained with human data?
2.)  The work claims to “significantly outperforms existing methods” (L11). Which one besides ACT?
3.) Can you elaborate on the design criteria of the new tasks? In what way do help to measure coordination?
4.)	How often did you run the experiments?

**Robotics Focus:**

4

**Summary Of Paper:**

This work extends the ACT framework by introducing a hierarchical attention mechanism for bimanual manipulation tasks, named InterACT. InterACT includes a Hierarchical Attention Encoder and a Multi-arm Decoder. The multi-arm decoder consists of a synchronization block to facilitate the coordination of both arms.  The method is evaluated both in simulation as well in real-world. Ablation studies confirm the need for component, such as the synchronization block.

**Summary Of Recommendation:**

Good work, some parts require clarification, e.g. if it has been trained with scripted data or with human demonstrations

---

### Decision · Program_Chairs · 2024-09-04

**Decision:**

Accept

**Comment:**

Strengths:
+ InterACT is a new approach to multi-arm coordination in bimanual robot-learning.
+ The approach is demonstrated on real-robot tasks.
+ The ablations confirm the contributions of the proposed changes to ACT.

Weaknesses:
- InterACT is only benchmarked against one baseline: ACT. Other visuomotor baselines like Diffusion Policy could be included.
- Some details on the evaluation setting are missing.
- The performance gains over ACT are marginal, given the added complexity of InterACT.

Post rebuttal:
While there were some concerns regarding the novelty of the approach, InterACT's approach is non-trivial and an interesting contribution to bimanual manipulation.